

# How the fear of COVID-19 changed the way we look at human faces

Giovanni Federico[1], Donatella Ferrante[2], Francesco Marcatto[2] and Maria Antonella Brandimonte[3]

[1] IRCCS SDN, Naples, Italy
[2] Department of Life Science, University of Trieste, Trieste, Italy
[3] Laboratory of Experimental Psychology, Suor Orsola Benincasa University, Naples, Italy

## ABSTRACT

Do we look at persons currently or previously affected by COVID-19 the same way as we do with healthy ones? In this eye-tracking study, we investigated how participants ($N$ = 54) looked at faces of individuals presented as "COVID-19 Free", "Sick with COVID-19", or "Recovered from COVID-19". Results showed that participants tend to look at the eyes of COVID-19-free faces longer than at those of both COVID-19-related faces. Crucially, we also found an increase of visual attention for the mouth of the COVID-19-related faces, possibly due to the threatening characterisation of such area as a transmission vehicle for SARS-CoV-2. Thus, by detailing how people dynamically changed the way of looking at faces as a function of the perceived risk of contagion, we provide the first evidence in the literature about the impact of the pandemic on the most basic level of social interaction.

## INTRODUCTION

The COVID-19 pandemic caused by SARS-CoV-2 is a global health concern that may cause severe threats to individuals' lives (*World Health Organization, 2020a*). Alongside the potentially fatal disease and the wide variability of physical health problems caused by the new beta-coronavirus, it is increasingly evident that the pandemic produced strong social, economic, and psychological effects (*Bavel et al., 2020*; *Brooks et al., 2020*; *Fiorillo & Gorwood, 2020*; *Pfefferbaum & North, 2020*). Therefore, in a fast and sadly predictable way, many clinical and psycho/sociological findings related to COVID-19's collateral effects have been accumulated. However, much less has been said about how the aftereffects of the current situation may reverberate on non-pathological, daily-life cognitive functioning. After all, if COVID-19 changed the world as we have been knowing it, one might wonder whether and, if so, how the "cognitive interface" between humans and the (post-COVID-19) world may transform.

When thinking about which "sensors" might primarily constitute such a kind of human interface with the world, the response is straightforward: Eyes. Indeed, studying how humans look at their surroundings may provide very useful insights about the cognitive processes underlying a wide variety of human behaviours as well as how people interact

Corresponding author
Giovanni Federico,
research@giovannifederico.net

with the environment (*Federico & Brandimonte, 2020*; *Hayhoe & Ballard, 2005*; *Henderson, 2017*; *Liversedge & Findlay, 2000*; *Milner & Goodale, 2006*; *Rahal & Fiedler, 2019*; *Rayner, 1998*). From the heterogeneous sets of stimuli present in the environment, there are some that mainly attract the attention of human beings when it comes to interacting with their own kind: others' human faces and, specifically, others' eyes (*Senju & Johnson, 2009a*). Indeed, more than nose and mouth, eyes constitute a major visual-attentional target for adults involved in face exploration (*Hernandez et al., 2009*; *Mertens, Siegmund & Grüsser, 1993*; *Walker-Smith, Gale & Findlay, 1977*). Also, humans tend to shift attention according to others' eye-gaze direction. Thus, as well as capturing and holding attention, eye-gaze may also elicit spatial orienting (for a comprehensive review, see *Dalmaso, Castelli & Galfano, 2020*).

Whereas in many species direct-gaze perception may produce an aversive response (*Emery, 2000*), the so-called "eye-contact effect"—by describing a broad set of neurocognitive effects associated with making eye contact with others—seems to be a foundation of human social interaction, hence constituting the basis of social cognition (*Csibra & Gergely, 2006*; *Kleinke, 1986*; *Senju, Johnson & Csibra, 2006*; *Senju & Johnson, 2009a*). In addition, in the context of self-referential processing (i.e., the way humans process stimuli concerning themselves), direct-gaze perception appears to produce a wide range of social-related effects. Indeed, the so-called "watching-eyes effects" may favour pro-social actions, positive appraisals of others, memory and self-awareness (*Conty, George & Hietanen, 2016*).

Although at different degrees of consciousness, eye-contact effects seem to be governed by the so-called "social brain", an extensive and composite brain network involved in human social interaction (*Adolphs, 2009*). To date, an atypical eye-contact pattern is one of the most significant symptoms of Autism Spectrum Disorder, which is a disorder that severely affects social functioning (*American Psychiatric Association, 2013*; *Senju & Johnson, 2009b*). Also, the intranasal administration of oxytocin, a neuropeptide mainly implicated in social-communicative function (*Insel, 2010*), appears to enhance eye contact in both autistic and neurotypical individuals (*Auyeung et al., 2015*). Thus, the mutual eye-contact experience we do daily when interacting with people probably substantiates one of the most powerful mechanisms to engage others (*Senju & Johnson, 2009b*). It seems, therefore, plausible that such a characteristic social behaviour might be affected by how severely COVID-19 impacted our psychosocial functioning, habits, and social interaction (*Bagcchi, 2020*; *Bavel et al., 2020*).

As stated above, most people fear the COVID-19 pandemic (*Pakpour & Griffiths, 2020*). Fear is a foundation emotion whose function is to protect animals against dangerous, threatening and aversive situations (*Misslin, 2003*). The neural counterparts of such an emotion are the cortical and subcortical areas implicated in the social brain network (*LeDoux, 2003*). Indeed, fear's brain networks are in charge of distinct defensive responses (e.g., flight, fight, freezing, avoidance) and may be triggered automatically either by unconditioned or conditioned stimuli (*Misslin, 2003*). In particular, the activity of the amygdala, an important crossroad of human emotional life, plays a significant role in reading social signals from the face, particularly in modulating direct gaze on others

(*Adolphs, 2010*; *Hietanen, 2018*; *Johnson et al., 2005*; *Kawashima et al., 1999*). This is true even for cortically blind patients (*Burra et al., 2013*). Thus, whereas others' direct gaze typically signal attention and social inclusion (*Wirth et al., 2010*), threatening stimuli, such as threat-priming faces, should be hardly considered as "social-engaging cues" so that one would likely avoid direct gaze with them (*Skuse, 2003*). In particular, if COVID-19-infected people were recognised as a possible threat to individual health, one might perceive them as minacious stimuli and, therefore, avoid or reduce direct eye-contact with them. In other words, others' human faces visual exploration might subtly change as an effect of the perceived risk of COVID-19 contagion.

Whereas making a specific experimental hypothesis about the lower duration of eye contact for COVID-19-related faces appears intriguing in itself, it may open the avenue to a further research question: how are other aspects of COVID-19-related faces looked at? Some useful insights might come by considering how threatening stimuli can automatically capture attention, even unconsciously (*Lin, Murray & Boynton, 2009*). Take, for example, the well-known "weapon focus" effect. The higher concentration of a crime eyewitness' attention on the weapon (i.e., the threatening stimulus) may result in a lower ability to remember other crime details (*Loftus, Loftus & Messo, 1987*). Thus, one may ask what kind of threat-related information can be "extracted" from COVID-19-related faces.

According to the World Health Organization, infected people may spread the SARS-CoV-2 by emitting small liquid particles (i.e., larger respiratory "droplets" or smaller "aerosols") through their mouth (*World Health Organization, 2020b*). Such a piece of critical information—which substantiates the effectiveness of face masks in preventing transmission of SARS-CoV-2 (*Bhardwaj & Agrawal, 2020*)—has been repeatedly reported by media, hence becoming part of the semantic knowledge about the post-COVID world. In a sense, the mouth seems to be an important transmission vehicle for the virus, so that one may reasonably predict that this area of a human face might represent a "threatening stimulus" for an interlocutor who, for some reasons, finds him/herself interacting with a person infected with COVID-19. Thus, whereas, on the one hand, one might expect lower eye contact for COVID-19-related faces, on the other hand, one may also predict an increase of attention for those face areas that are typically associated with the risk of virus transmission (i.e., the mouth).

To test the above hypotheses, in the present study, we analysed by eye-tracking the visual-attentional patterns of participants engaged in an online, ecological, free-observation task in which they were simply required to look at human-face stimuli generated by Artificial Intelligence. We manipulated the perceived risk of infection prompted by the faces by randomly indicating the immunological status of each face as "COVID-19 free" (i.e., individuals who never contracted the virus), "Sick with COVID-19" (i.e., individuals who are currently infected with COVID-19), or "Recovered from COVID-19" (i.e., individuals who got COVID-19 but who have now fully recovered). We included the "Recovered from COVID-19" condition to assess whether the COVID-19-related stigma (*Bagcchi, 2020*) may reverberate on the way people look at faces of patients who survived COVID-19. We thereby conjectured that participants

should tend to avoid eye contact for individuals presented as suffering from COVID-19. Also, COVID-19-related faces should attract participants' visuospatial attention towards the area of the stimulus implicitly recognised as threatening, that is, the mouth. Finally, we included *ad-hoc* self-report psychometric measures (i.e., a post-experimental interview; Supplemental Material 1) to assess participants' perceived risk of contagion in relation to their attitude towards the COVID-19 pandemic.

## MATERIALS & METHODS

Due to the pandemic situation at the study's date (November 2020), we devised a web-based, online experiment by using custom software and scripts. All participants were safe at home whilst participating in the study and used their own devices (i.e., personal computers or notebooks) to perform the experiment. All experimental procedures followed the ethical standards laid down in the Declaration of Helsinki (1964). Accordingly, the study received approval (approval number: CVD-19-ET) from the Ethics Committee of Suor Orsola Benincasa University (Naples, Italy).

### Participants

Fifty-four participants (31 females; mean age = 26.46 years, SD = 5.82) with self-reported normal or corrected-to-normal vision were enrolled in the experiment. Participants were all Caucasians. We calculated the sample size on the basis of previous similar studies (*Hernandez et al., 2009*; *Mertens, Siegmund & Grüsser, 1993*; *Walker-Smith, Gale & Findlay, 1977*) and by considering an a-priori power analysis (*Cohen, 2013*; *Faul et al., 2007*) to detect a small effect size ($\eta_p^2 = 0.20$) within a repeated-measures ANOVA, with a power of 0.90 and an alpha level of 0.05 (computed $N = 53$). All participants had no history of neurological or psychiatric disorders and gave informed consent on their participation by indicating their explicit consent via a specific online form. Four female participants were excluded from data analyses due to their performance above 3.0 SD (outliers).

### Materials

In this study, we used images of human faces generated by Artificial Intelligence and an ad-hoc post-experimental interview.

### Human faces

For the experiment, we used 18 images of faces (9 females) generated through machine learning by implementing a generative adversarial network (*Karras et al., 2020*).
By adopting GAN-generated faces, we kept constant the aspect ratio between eyes and mouth, hence maintaining invariant the spatial disposition and size of all anatomical features of the faces. Once images were generated, we erased each one's background by using a proprietary algorithm freely available online at the URL "https://www.remove.bg" (Kaleido AI GmbH, Austria). Then, we grayscaled all the images using the specific function of the KRITA open-source raster graphics editor (v.4.4.1 for Apple macOS). As a result,

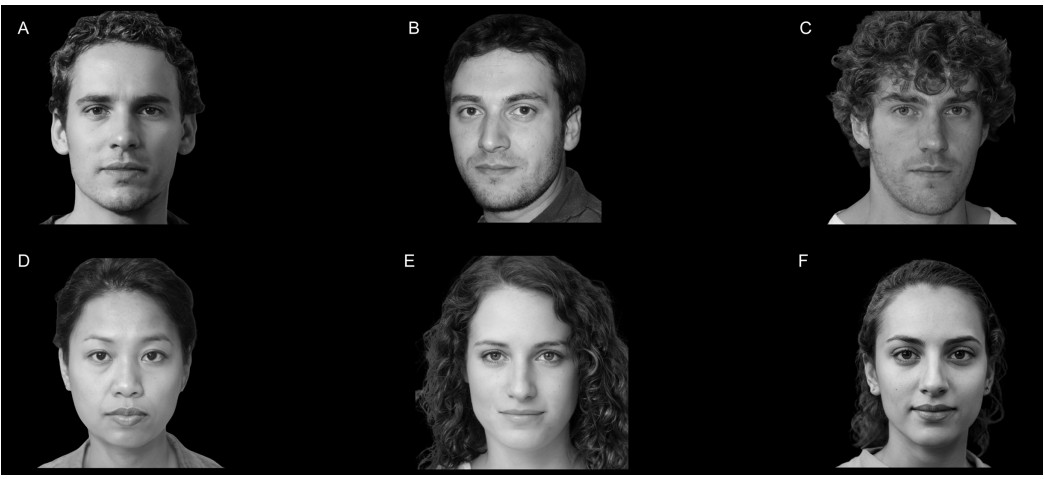

**Figure 1 Example of stimuli used in the study.** Example of faces used in the experiment (A–F). All the faces were generated through machine learning by implementing a generative adversarial network (*Karras et al., 2020*).

we obtained 18 images of monochrome faces that appear as real to the human eye. Before enrolling participants and starting with the study, we assessed the degree of authenticity of all the GAN-generated faces by asking independent testers ($N = 10$; 5 females) to check whether each face appeared real. All the independent testers considered all the stimuli as real faces. An example of faces involved in the study can be seen in Fig. 1. We randomly divided the faces into three groups that correspond to the three experimental conditions of the experiment: 6 × "COVID-19 Free", 6 × "Sick with COVID-19", 6 × "Recovered from COVID-19". Each experimental condition was matched for face sex (3 females for each condition). We changed the experimental-condition assignment for each group of faces at every 18 participants. Thus, at the end of the study ($N = 54$), each face was displayed in all the experimental conditions, hence controlling for possible effects generated by the specific salience of individual stimuli.

## Post-experimental interview

We developed an ad-hoc post-experimental interview (PEI) for the study (Supplemental Material 1). The PEI was introduced to assess participants' risk perception of coming into contact with a COVID-19 patient as well as their propensity to engage in protective behaviours. The risk perception items included six risk judgments adapted from prior research on the psychometric paradigm (*Slovic, 1987*, *2016*), which evaluated the following facets of perceived risk: probability, fear, knowledge, control, and severity. Regarding protective behaviours, participants were asked to evaluate how often they engaged in COVID-19-protective behaviours (such as using the hand sanitiser) in the previous two weeks (6 items) and how often they plan to engage in COVID-19-protective behaviours in the next two weeks (6 items). The PEI also included demographic information (7 items) and questions about prior exposure to COVID-19 (4 items). The PEI was edited and published online by using the Google Forms platform.

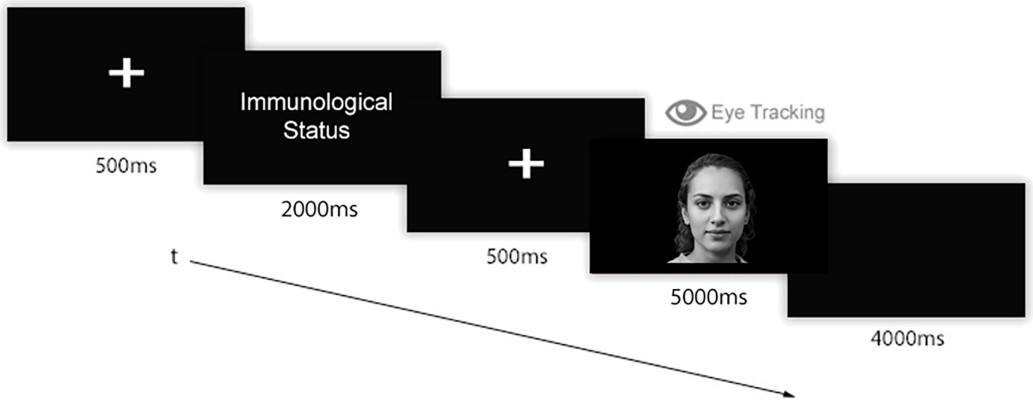

**Figure 2 Experimental flow.** A fixation point appeared for 500 ms, followed by a label indicating the Immunological Status (i.e., "COVID-19 free", "Sick with COVID-19", or "Recovered from COVID-19"), which was shown for 2,000 ms. Then, a second fixation point (500 ms) appeared followed by a face stimulus that remained on the screen for 5,000 ms. Finally, a black screen appeared for 4,000 ms.

## Procedure

Participants received a link through which they could access the online platform for the experiment. Once accessed and prior to the test, participants digitally signed informed consent. Instructions about the experiment and the required experimental setting appeared on the screen. Participants were asked to configure their webcam by following the instruction they read on screen. Then, they completed an eye-tracking calibration procedure by looking at and then clicking on red dots that sequentially appeared on 40 parts of the screen. Afterwards, the eye-tracking study started. The experimental instructions were: "*Now you will see some people's faces. Before each face, you will read if the person is currently SICK WITH COVID, RECOVERED FROM COVID, or NEVER SICK WITH COVID. Please, look at faces in the most natural way possible*". Then, the eye-tracking experiment started. A single trial of 6 images of faces related to each experimental condition was administered. Thus, 18 faces were randomly presented according to the experimental visual flow (Fig. 2): before each stimulus, a fixation point (i.e., a screen-centred white cross over grey background) was shown for 500 ms. Then, a label indicating the state of health of the coming face appeared for 2,000 ms, followed by a second fixation point (500 ms). Then, a face appeared for 5,000 ms. After the face, a black screen appeared for 4,000 ms in order to permit the retina to relax. Each single presentation lasted 12 s. Overall, the stimuli presentation lasted 216 s. At the end of the stimulation, participants were redirected to a Google Forms webpage to conclude the experiment by responding to the Post Experimental Interview (PEI). At the end of the PEI, participants were asked to evaluate the authenticity of the face stimuli they had previously seen. Globally, the study lasted about 20 min for each participant. At the end of the experiment, participants were debriefed regarding the study's purposes and the methods by redirecting them to a specific webpage. All participants reported all stimuli as being real human faces.

## Apparatus and software paradigms

Participants used their devices to access the online experiment. Technical prerequisites for participating in the study were using Google Chrome browser and having a personal computer or a notebook with a webcam. We developed the online experiment by using different classes of technologies. For the user-interaction interface (e.g., experimental instructions page, page transitions, etc.), we developed webpages using PHP programming language, JavaScript script language, and MySQL database. Those pages acted as a bridge between the different parts of the experiment, hence guiding participants from the first (i.e., eye-tracking part), to the second (i.e., post-experimental interview), and then, to the third part of the study (i.e., participants' final evaluation of the stimuli). To acquire participants' gaze data, we used the RealEye.io (RealEye sp. z o.o., Poland) platform, an online eye-tracking technology based on WebGazer library (*Papoutsaki et al., 2016*). The eye-tracking technology we used (*Papoutsaki et al., 2016*) has been compared with other commercial-grade, high-level eye-tracking systems and sensors (*Semmelmann & Weigelt, 2018*). When analyses do not require a very detailed spatial resolution, as in the detail level required by this study, results between systems appear to be comparable. Thus, online webcam-based eye tracking has been proved to be a reliable solution in such a kind of cognitive studies (*Semmelmann & Weigelt, 2018*). We included in the study only participants who had devices and webcams capable of obtaining at least a sampling rate of 20 Hz. To construct and publish the post-experimental interview, we used the Google Forms platform. To extract and analyse participants' gaze data, we engineered and developed different ad-hoc, custom-made scripts using PHP programming language and the MySQL Database Management System. All the face stimuli involved in the experiment were presented at the best-fitted resolution for participants' displays (auto-resizing stimuli).

## Gaze-behavioural data

We analysed gaze-behavioural data in terms of dwell time, that is, the amount of time (expressed in milliseconds) that participants spent looking at different Areas of Interest (AOIs). We thereby defined two distinct AOIs: the eye area (i.e., a rectangular area comprising both the eyes) and the mouth area (i.e., a rectangular area including the mouth). For all the stimuli, we maintained fixed both the size and the spatial position of the AOIs. An example of the AOIs used in the study can be seen in Fig. 3. For the eye-tracking data analyses, we used a two-time-window approach (*Federico & Brandimonte, 2019*) aimed at studying the initial (i.e., the first 500 ms) and the full visual exploration of the faces (i.e., all the 5,000 ms). We chose a two-time approach to characterise the time course of participants' visual-spatial exploration. Therefore, we analysed the first-500 ms time window to explore the initial stage of participants' visual exploration. Such a time interval acted as an at-a-first-glance indication in data analysis, highlighting participants' initial fixation patterns as soon as the stimuli appeared. Secondly, we extended the time window of analysis by including participants' full visual exploration (i.e., 5,000 ms), thus taking into account the effect of time on the experimental manipulations we made. A preliminary

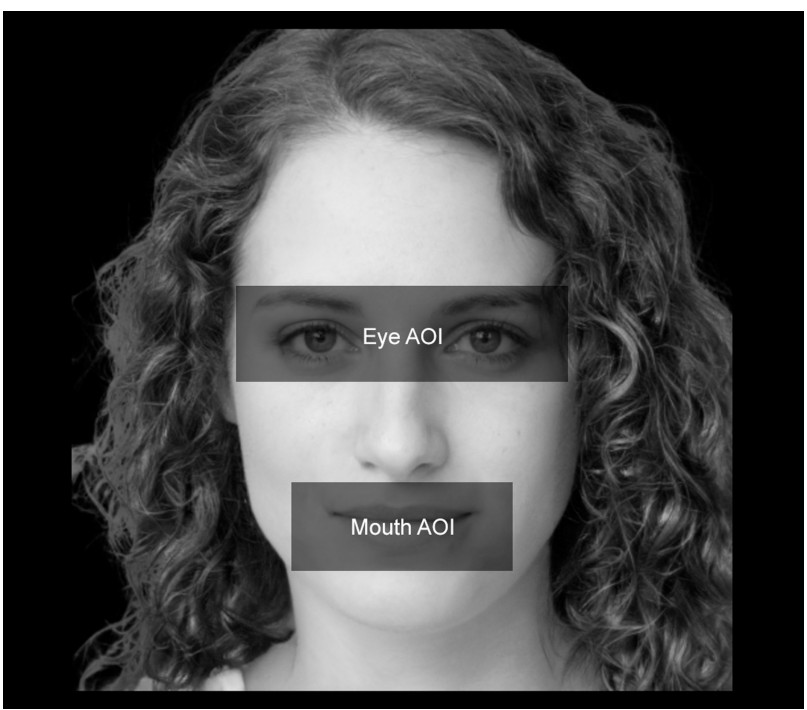

**Figure 3 Face AOIs considered in the study.** We analysed how participants looked at two distinct AOIs of the faces involved in the study. The first AOI was associated with the eyes (i.e., "Eye AOI"). The second AOI referred to the mouth (i.e., "Mouth AOI"). Both AOIs remained stable in terms of size, spatial position, and proportions across the stimuli. 

qualitative indication of differences in participants' visual-attentional patterns can be appreciated in the heatmaps depicted in Fig. 4.

## Data analyses

We implemented multiple data analyses. First, we considered how participants looked at the faces as the information we manipulated about their immunological status changed. Thus, we performed two $3 \times 2$ repeated measure analysis of variance with "Immunological Status" ("COVID-19-free" vs. "Sick with COVID-19" vs. "Recovered from COVID-19") as a 3-level factor and "AOIs" (eyes vs. mouth) as a 2-level factor on face fixation duration (expressed in milliseconds) for each time window of analysis (500 ms and 5,000 ms). Secondly, we analysed the post-experimental interviews (PEI) data to explore whether and, if so, to what extent participants' risk perception towards COVID-19 and COVID-19-related behaviours and intentions were associated with their gaze-behavioural data. Risk perception is known to be a multi-dimensional construct (*Slovic, 1987*); therefore, we conducted a principal component analysis with varimax rotation to investigate its dimensional structure. Data were previously checked for sphericity and sampling adequacy using the Kaiser–Meyer–Olkin (KMO) and Bartlett tests. The other constructs involved in the PEI were expected to be mono-dimensional, hence we aggregated the responses to obtain, for each participant, a COVID-19-related behaviour score and a COVID-19-related intention score. Pearson correlations were then

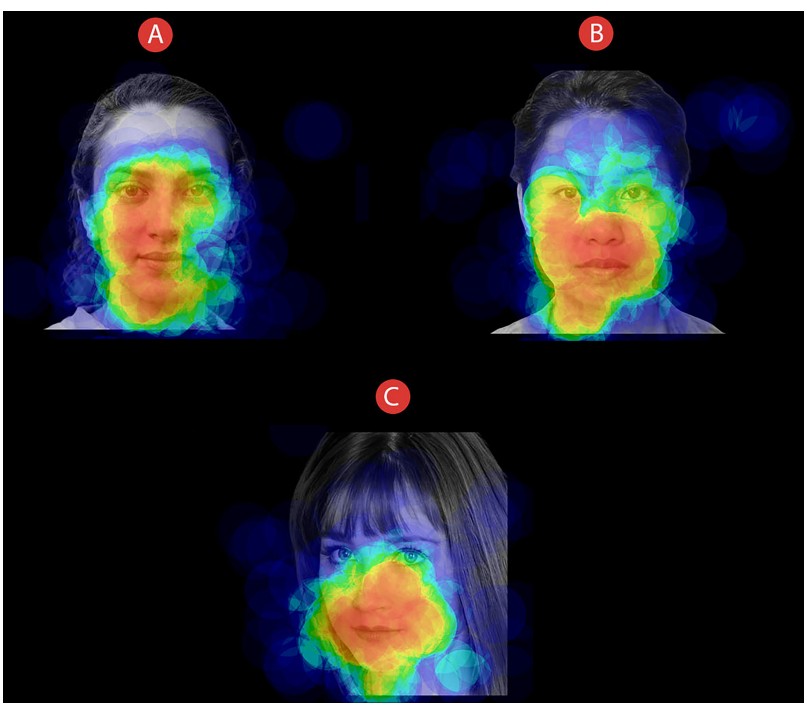

**Figure 4 Face exploration heatmaps.** Example of heatmaps related to how all participants explored faces of individuals that could be presented as (A) COVID-19 free, (B) Sick with COVID-19, or (C) Recovered from COVID-19. Time window: 5,000 ms.

**Table 1 Initial visual exploration (500 ms).** Eye-tracking data (sampling rate: 20 Hz) related to participants' initial visual exploration.

| Immunological status | AOIs–Dwell time (SD) | |
| --- | --- | --- |
| | **Eyes** | **Mouth** |
| COVID-19 free | 129.26 (86.06) ms | 74.97 (59.44) ms |
| Sick with COVID-19 | 105.86 (80.4) ms | 76.23 (56.11) ms |
| Recovered from COVID-19 | 105.71 (83.36) ms | 84.96 (68.19) ms |

calculated between the PEI scales and the face fixation duration in each condition, both at 500 ms and 5,000 ms. We used an alpha level of 0.05 for all the analyses and Tukey HSD correction for the post-hoc comparisons.

## RESULTS

### Eye-tracking results

Eye-tracking data related to participants' initial visual exploration (i.e., 500 ms) are summarised in Table 1, whereas data related to the full exploration (i.e., 5,000 ms) are reported in Table 2.

### *Initial visual exploration (first 500 ms)*

Regarding the way participants initially visually explored the faces (i.e., within the first 500 ms), the results of a repeated-measure analysis of variance showed an interaction

**Table 2 Full visual exploration (5,000 ms).** Eye-tracking data (sampling rate: 20 Hz) related to participants' full visual exploration.

| Immunological status | AOIs – Dwell Time (SD) | |
| --- | --- | --- |
| | Eyes | Mouth |
| COVID-19 free | 1,191.48 (790.63) ms | 678.25 (403.02) ms |
| Sick with COVID-19 | 991.99 (670.22) ms | 859.96 (391.528) ms |
| Recovered from COVID-19 | 982.39 (710.69) ms | 864.87 (424.02) ms |

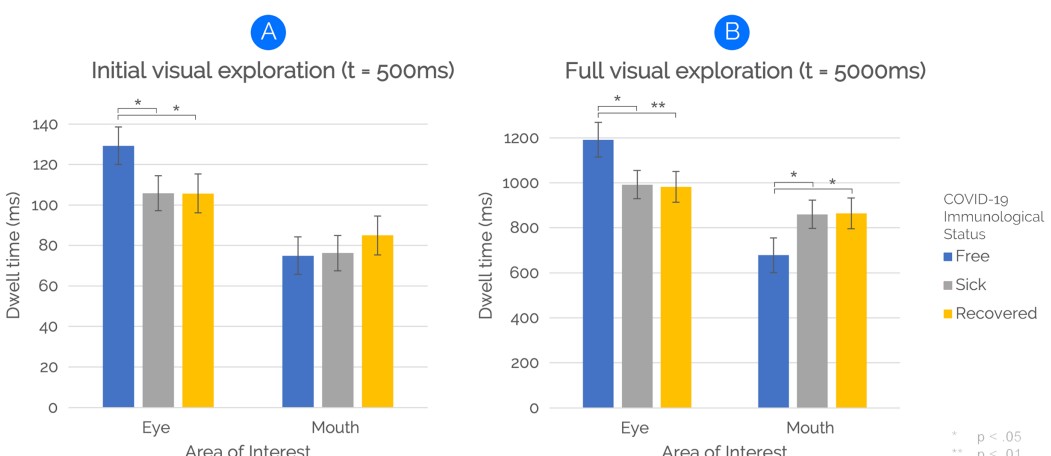

**Figure 5 The effect of COVID-19 Immunological Status on face visual exploration.** Participants looked at the eyes of individuals presented as "COVID-19 Free" longer than at those of individuals presented as "Sick with COVID-19" or "Recovered from COVID-19" for both time windows of analysis (A and B). Additionally, the full exploration analysis (B) revealed that mouth areas of both sick-with-COVID-19 and recovered-from-COVID-19 faces were fixated longer than COVID-19-free ones. Vertical bars denote 0.95 confidence intervals, computed by adopting a simpler solution to *Loftus & Masson (1994)* provided by *Cousineau (2005)*.

between "Immunological Status" and "AOIs", $F(2, 98) = 3.01$, $p = 0.023$, $\eta^2_p = 0.074$. Post-hoc pairwise comparisons revealed that eyes of COVID-19-free faces were fixated longer than eyes of COVID-19-related (i.e., "Sick with COVID" and "Recovered from COVID") faces ($p < 0.05$). The interaction effect is shown in Fig. 5A. No main effects of "Immunological Status" or "AOIs" were found.

### Full visual exploration (5,000 ms)

By extending the time window of eye-tracking analysis to 5,000 ms to examine participants' full visual exploration, the results of a repeated-measure analysis of variance revealed a main effect of "AOIs" on face visual exploration, $F(1,49) = 4.19$, $p = 0.046$, $\eta^2_p = 0.079$. The main effect was due to longer fixations to Eye AOIs than Mouth AOIs. However, and most important, an interaction between "Immunological Status" and "AOIs" was also found, $F(2,98)$, $p < 0.001$, $\eta^2_p = 0.196$. Post-hoc pairwise comparisons revealed that eyes of COVID-19-free faces were fixated longer than eyes of sick-with-COVID-19 faces ($p = 0.013$) and recovered-from-COVID-19 faces ($p < 0.01$). Also, the

**Table 3 Post-experimental Interview results.**

| Item | | |
|---|---|---|
| **COVID-19-related information** | **Yes (%)** | **No (%)** |
| *2.1 I am currently COVID-19 positive* | 1 (2) | 48 (98) |
| *2.2 I got sick of COVID-19, but now I'm cured and negative* | 3 (6.1) | 46 (93.9) |
| *2.3 In my circle of acquaintances, someone got sick of COVID-19* | 42 (85.7) | 7 (14.3) |
| *2.4 In my circle of acquaintances, someone died from COVID-19* | 14 (28.6) | 35 (71.4) |
| **Risk perception** | **M** | **SD** |
| *3.1 How risky is it for you coming into contact with a COVID-19 patient?* | 5.45 | 1.51 |
| *3.2 How likely is it for you coming into contact with a COVID-19 patient?* | 5.06 | 1.45 |
| *3.3 How much can you control the possibility of coming into contact with a COVID-19 patient?* | 4.27 | 1.30 |
| *3.4 Are you afraid of coming into contact with a COVID-19 patient?* | 4.82 | 1.72 |
| *3.5 Do you know what the consequences of coming into contact with a COVID-19 patient are?* | 6.33 | 0.92 |
| *3.6 How serious might be for you the consequences of coming into contact with a COVID-19 patient?* | 5.43 | 1.17 |
| **COVID-19-related behaviours** | | |
| *4.1 Used the mask* | 5.86 | .35 |
| *4.2 Used gloves* | 2.06 | 1.25 |
| *4.3 Used the hand sanitizer* | 5.18 | 1.03 |
| *4.4 Kept a distance of at least one meter from other people* | 5.06 | .92 |
| *4.5 Avoided crowded places* | 5.33 | 0.80 |
| *4.6 Avoided meeting friends/relatives* | 4.41 | 1.35 |
| **COVID-19-related intentions** | | |
| *5.1 Use the mask* | 5.94 | 0.32 |
| *5.2 Use gloves* | 2.63 | 1.32 |
| *5.3 Use the hand sanitizer* | 5.47 | 0.77 |
| *5.4 Keep a distance of at least one meter from other people* | 5.33 | 0.80 |
| *5.5 Avoid crowded places* | 5.53 | 0.68 |
| *5.6 Avoid meeting friends/relatives* | 4.39 | 1.10 |

mouth was fixated longer for both sick-with-COVID-19 ($p = 0.032$) and recovered-from-COVID-19 ($p = 0.025$) faces than for COVID-19-free faces. The interaction effect is shown in Fig. 5B. No main effect of "Immunological Status" was found.

### Post-experimental interview

PEI responses are summarised in Table 3. A principal component analysis was conducted to assess the dimensional structure of the six risk-perception items. Preliminarily, we checked data for sampling adequacy and sphericity: the KMO value was 0.59 and Bartlett's test of sphericity was significant ($\chi^2(15) = 44.32$, $p = 0.001$), thus indicating that the sampling was adequate and the inter-item correlations were large enough, hence suggesting that a principal component analysis was appropriate. The number of factors retained was determined through parallel analysis (*Dinno, 2009*; *Horn, 1965*), inspection of scree plots and interpretability of each component. The parallel analysis suggested a two-factor solution accounting for 58% of the total variance, but both the scree plot and the components' interpretability supported instead the extraction of three factors, which

**Table 4 Risk perception factor analysis.**

| | Components | | |
|---|---|---|---|
| Risk perception items | 1 | 2 | 3 |
| *3.1* | 0.87 | 0.09 | −0.04 |
| *3.4* | 0.76 | −0.18 | 0.17 |
| *3.6* | 0.68 | 0.48 | 0.11 |
| *3.3* | 0.13 | −0.80 | 0.22 |
| *3.2* | 0.15 | 0.76 | 0.12 |
| *3.5* | 0.09 | −0.03 | 0.98 |

Note:
Risk perception factor analysis (extraction method: principal component analysis with varimax rotation). Component 1 has been named *Severity*, Component 2 *Probability* and Component 3 *Knowledge*.

**Table 5 Pearson's correlation coefficients among the PEI scales and gaze-behavioural data.**

| Item | Severity | Probability | Knowledge | COVID-19-related behaviours | COVID-19-related intentions |
|---|---|---|---|---|---|
| Severity | – | | | | |
| Probability | 0.13 | – | | | |
| Knowledge | 0.19 | −0.09 | – | | |
| COVID-19-related behaviours | 0.17 | −0.02 | 0.27 | – | |
| COVID-19-related intentions | 0.23 | −0.01 | 0.27 | 0.84** | – |
| Dwell Time-COVID-19-free eyes 500 ms | −0.29* | −0.06 | −0.03 | −0.22 | −0.13 |
| Dwell Time-COVID-19-free mouth 500 ms | 0.22 | 0.03 | 0.02 | −0.06 | 0.00 |
| Dwell Time-COVID-19 sick eyes 500 ms | −0.19 | 0.03 | −0.09 | −0.14 | −0.06 |
| Dwell Time-COVID-19 sick mouth 500 ms | 0.15 | −0.01 | −0.05 | −0.22 | −0.08 |
| Dwell Time-COVID-19 recovered eyes 500 ms | −0.08 | 0.03 | 0.02 | −0.03 | 0.07 |
| Dwell Time-COVID-19 recovered mouth 500 ms | 0.18 | 0.01 | −0.06 | −0.23 | −0.17 |
| Dwell Time-COVID-19-free eyes 5,000 ms | −0.17 | −0.09 | −0.01 | −0.17 | −0.07 |
| Dwell Time-COVID-19-free mouth 5,000 ms | 0.18 | 0.11 | −0.13 | −0.06 | 0.02 |
| Dwell Time-COVID-19 sick eyes 5,000 ms | -0.14 | 0.03 | 0.07 | 0.01 | 0.11 |
| Dwell Time-COVID-19 sick mouth 5,000 ms | 0.04 | 0.04 | −0.04 | −0.11 | −0.02 |
| Dwell Time-COVID-19 recovered eyes 5,000 ms | −0.10 | −0.08 | −0.05 | −0.02 | 0.07 |
| Dwell Time-COVID-19 recovered mouth 5,000 ms | 0.41** | 0.07 | 0.08 | −0.13 | −0.10 |

Notes:
* $p < 0.05$.
** $p < 0.01$.

accounted for 73% of the total variance (Table 4). Items 3.1, 3.4, and 3.6 represented the *Severity* dimension (Cronbach's $\alpha = 0.68$), items 3.2 and 3.3 represented the *Probability* dimension (Cronbach's $\alpha = 0.45$), and item 3.5 the *Knowledge* dimension. Scores of Sections 4 and 5 items were aggregated separately to create a COVID-19-related behaviour scale (Cronbach's $\alpha = 0.60$) and a COVID-19-related intention scale (Cronbach's $\alpha = 0.71$), respectively. Pearson's correlation coefficients among the PEI scales and the gaze-behavioural data are reported in Table 5. An interesting pattern emerged, with face

fixation duration and scores in the perceived Severity being negatively correlated with fixations to Eye AOIs and positively correlated with fixations to Mouth AOIs (both at 500 ms and 5,000 ms), although the only statistically significant correlations were with the Mouth AOI in the recovered-from-COVID-19 condition (at 5,000 ms, $r = 0.41$, $p = 0.003$) and with the Eye AOI in the COVID-19-free condition (at 500 ms, $r = -0.29$, $p = 0.043$). No significant correlations were found with the other dimensions of risk perception, nor with the COVID-19-related behaviour and intention scales.

## DISCUSSION

In this study, we examined how participants looked at faces of individuals who could be presented as "COVID-19 Free", "Sick with COVID-19", or "Recovered from COVID-19". In so doing, we aimed at investigating how the multitude of psychosocial effects produced by the current pandemic (*Bagcchi, 2020*; *Bavel et al., 2020*) and the related state of generalised apprehension and fear (*Pakpour & Griffiths, 2020*) may reverberate on daily-life, socially-based human cognitive functioning. Specifically, we chose to analyse how the current dramatic contingencies may affect the way people interact with one another. We focused our attention on how humans look at others' faces because *eye-contact effects* constitute one of the most powerful and human-characterising social behaviours (*Csibra & Gergely, 2006*; *Hernandez et al., 2009*; *Kleinke, 1986*; *Mertens, Siegmund & Grüsser, 1993*; *Senju & Johnson, 2009a*; *Walker-Smith, Gale & Findlay, 1977*; *Wirth et al., 2010*; *Conty, George & Hietanen, 2016*; *Dalmaso, Castelli & Galfano, 2020*). Therefore, we posited that faces of people who were considered possible threatening stimuli due to their COVID-19-related illness might become the target of distinctive visual-attention patterns as compared to the more reassuring faces of COVID-19-free individuals, possibly because COVID-19-related faces activate implicit cognitive mechanisms associated with risk avoidance and fear (*Adolphs, 2009*, *2010*; *Hietanen, 2018*; *Johnson et al., 2005*; *Kawashima et al., 1999*; *LeDoux, 2003*; *Lin, Murray & Boynton, 2009*; *Loftus, Loftus & Messo, 1987*; *Misslin, 2003*; *Skuse, 2003*).

Consistent with our predictions, participants looked at the eyes of faces presented as "COVID-19 Free" significantly longer than faces presented as "Sick with COVID-19" or "Recovered from COVID-19". Such a peculiar visual-attentional pattern seems to emerge from the very first visual exploration time interval (i.e., 500 ms), suggesting, as an at-first-glance indication, an initial detachment-from-threatening-stimuli mechanism (see Fig. 5A). However, and most important, by extending the time window of the eye-tracking analysis to 5,000 ms—thus covering participants' full visual exploration—we found the same higher eye-focused pattern for COVID-19-free faces. Notably, within the extended time window, the main effect of *AOIs* with eyes receiving longer fixations than mouth was mitigated by a significant interaction. Thus, when considering both time windows of analysis, results clearly indicate that participants' implicit visual-attention patterns were modulated by explicit information on the health status of to–look-at faces, with a higher amount of eye contact for COVID-19-free faces than for both sick-with/recovered-from-COVID-19 faces.

Along with differences in eye contact as a function of health status, within the extended time window of analysis (5,000 ms), we found a longer time allocation of visual-spatial attention for the mouth's face area under both sick-with-COVID-19 and recovered-from-COVID-19 conditions as compared to the COVID-19-free condition. Interestingly, participants visually explored the eyes and the mouth of COVID-19-related faces in a complementary way. Indeed, participants looked at the eyes of COVID-19-related faces about 200 ms less than COVID-19-free faces. Symmetrically, participants fixated the mouth of COVID-19-related faces about 200 ms longer than COVID-19-free ones (see Fig. 5B). Thus, besides the implicit and sudden detachment-from-eyes mechanism, shorter eye fixation strongly supports the idea of a greater attraction exerted by the mouth in the quality of the threat-related area of the stimulus (*Lin, Murray & Boynton, 2009*; *Loftus, Loftus & Messo, 1987*). Indeed, as we stated above, SARS-CoV-2 may spread through respiratory droplets and aerosols (*World Health Organization, 2020b*). Such contagion-related information is nowadays part of the semantic reservoir of the general population. Therefore, one may reasonably assume how the threatening characterisation of the mouth may depend on the specific transmission modality of the virus (i.e., airborne). Whereas this study represents the first exploration of such a complex phenomenon in the context of the COVID-19 pandemic, future studies should explicitly explore the risk-perception mechanisms associated with other diseases.

Critically, we found overlapping results for sick-with-COVID-19 and recovered-from-COVID-19 conditions for all the effects we found. Whereas such overlap might reflect the stigma associated with COVID-19 (*Bagcchi, 2020*), it may also depend on the compromised and uncertainty-governed informative context generated by the pandemic (*Koffman et al., 2020*). Indeed, the massive profusion of often inaccurate, or even fake, news about the disease outcomes and the transmission modalities may have led people to take an incorrect, anxiety-modulated precautionary attitude towards the individuals recovered from COVID-19 (*Usher et al., 2020*; *van der Linden, Roozenbeek & Compton, 2020*). In this regard, it should be noticed that individuals largely use Internet and social media to obtain information regarding COVID-19. The ability to check for the correctness of Internet-distributed information is inevitably limited so that the risk to get fake news is consistently high (*Lazer et al., 2018*). Also, the large availability of COVID-19-related information may generate information overload and overconcern among people, thus fuelling the fear of COVID-19 (*Farooq, Laato & Islam, 2020*).

Individuals are frightened by COVID-19 (*Pakpour & Griffiths, 2020*). To date, the fear of COVID-19 has led some individuals to commit suicide (*Goyal et al., 2020*; *Mamun & Griffiths, 2020*) and the size of the problem is so large that some scholars have built ad-hoc scales (*Ahorsu et al., 2020*). Based on that knowledge, besides gaze-behavioural measures, we included some *ad-hoc* psychometric measures to explore the perceived risk of COVID-19 contagion. The factorial analysis of participants' post-experimental-interview (PEI) revealed three risk-related main factors (i.e., perceived severity of COVID-19 contagion, probability of controlling the risk of infection, and knowledge about COVID-19) that explain over 73% of variance. This result is consistent with the typical factorial structures reported in the risk-perception literature, which include both cognitive

(knowledge and probability of controlling the risk) and emotional (severity) dimensions (*Loewenstein et al., 2001*; *Oh, Paek & Hove, 2015*; *Slovic, 1987*, *2016*). The idea that fear might be at the root of the overlapping results for sick-with-COVID-19 and recovered-from-COVID-19 conditions seems to be further supported by the correlation ($r = 0.41$) between perceived severity and time of observation of the Mouth AOIs of recovered-from-COVID-19 faces. Also, within our sample, the perceived severity of COVID-19 contagion was negatively correlated with fixations to the eyes and positively correlated with fixations to the mouth, within both time windows of analysis. However, it should be noted that, although adequate for studying eye movements, this study's sample size is too small to measure individual psychosocial differences. Therefore, the correlation analyses we propose here should be considered only as a non-exhaustive support to interpret and discuss the gaze-behavioural effects we found, which constitute the study's main result. Thus, by underlining how the most critical findings of the present research are those related to how participants changed their visual-attentional patterns as a function of faces' COVID-19 immunological status, the PEI data should be considered as potentially fruitful direction for future research aimed at exploring the most profound psychosocial implications of COVID-19.

To sum up, the interaction we found between immunological status and how participants looked at different characteristics of the faces suggests that the COVID-19-related contingencies we are experiencing may resonate on basic cognitive processing underlying social interaction. Indeed, in our study, participants' performance in an implicit task (free face observation) was substantially influenced by explicit information (immunological status) provided prior to the test. Persuaded by the evidence accumulated during the last twenty years of neuroscientific research, we are keen to interpret our results in terms of abrupt triggering of the neurocognitive systems involved in social functioning, fear and gaze-behavioural control (*Adolphs, 2009*, *2010*; *Hietanen, 2018*; *Johnson et al., 2005*; *LeDoux, 2003*; *Lin, Murray & Boynton, 2009*; *Loftus, Loftus & Messo, 1987*; *Misslin, 2003*; *Senju & Johnson, 2009a*; *Skuse, 2003*; *Wirth et al., 2010*). By providing the first evidence about the effects of the pandemic on the most basic level of social cognition (i.e., eye contact), the present research shed new light on how flexible and adaptive cognitive processing may lead humans to interact with the environment in a plastic way, by integrating multiple sources of information (*Federico, Osiurak & Brandimonte, 2021*; *Federico & Brandimonte, 2020*).

## CONCLUSIONS

COVID-19 pandemic produced and is still producing strong psychosocial effects within the general population. Whereas current research has addressed the clinical and bio-psycho-sociological effects of COVID-19, much less space has been devoted to the pandemic's consequences on non-pathological and daily-life cognitive functions. In the context of the neuroscientific/psychological perspective of social cognition, we investigated how humans modified their social-interaction modalities due to the COVID-19-related contingencies. In particular, we analysed how people looked at faces of individuals presented as COVID-19 free, sick with COVID-19, or recovered from COVID-19.

We found that participants tended to look at the eyes of COVID-19-free faces longer than at those of both COVID-19-related faces. Crucially, under both COVID-19-related conditions, the implicit detachment-from-eyes mechanism we report seems to be compensated by increasing visual attention to the mouth area. This increase suggests a threatening characterisation of the mouth as a transmission vehicle for SARS-CoV-2. Notably, such an implicit gaze-behavioural pattern appears to be consistent with the self-report psychometric measures we introduced to find out how participants perceived the risk of COVID-19 contagion. As an initial exploration of a very complex reality, this article reports the first evidence in the literature about the pandemic's psychological and social reverberations on the most basic level of human social interaction.

### Funding
The authors received no funding for this work.

### Competing Interests
Giovanni Federico is employed by IRCCS SDN.

### Author Contributions
- Giovanni Federico conceived and designed the experiment, performed the experiment, analyzed the data, prepared figures and/or tables, authored or reviewed drafts of the paper, and approved the final draft.
- Donatella Ferrante analyzed the data, authored or reviewed drafts of the paper, and approved the final draft.
- Francesco Marcatto analyzed the data, prepared figures and/or tables, authored or reviewed drafts of the paper, and approved the final draft.
- Maria Antonella Brandimonte conceived and designed the experiment, analyzed the data, authored or reviewed drafts of the paper, and approved the final draft.

### Human Ethics
The following information was supplied relating to ethical approvals (i.e., approving body and any reference numbers):

The Ethics Committee of Suor Orsola Benincasa University granted ethical approval to carry out the study (cvd-19-et).

### Data Availability
The raw data are available in the Supplemental Files.

### Supplemental Information
Supplemental information for this article can be found online at http://dx.doi.org/10.7717/peerj.11380#supplemental-information.

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
