# Peer review of "How the fear of COVID-19 changed the way we look at human faces"

_PeerJ, doi:10.7717/peerj.11380_

## Round 0.1 · original submission · Major Revisions

Dear Authors,

Please revised the manuscript as per comments from the three peer reviewers.

Reviewer 1 ·

Basic reporting

please see below

Experimental design

please see below

Validity of the findings

please see below

Additional comments

This is a nice study showing how covid-related information can shape the exploration of face stimuli. Briefly, participants made more fixations towards the eye region of faces described as belonging to covid-free individuals. Some comments are following.

1) As far as I have known, ”eye contact effect” is an expression introduced by Senju et al (Senju & Johnson, 2009) to describe a wide set of neuro-cognitive consequences of making eye contact with another person (hence, it is not only restricted to attentional mechanisms). Other research groups, such as that of Hietanen, provided further insights and proposed the “watching eyes effect” (Conty, George, & Hietanen, 2016). You could therefore provide a short overview of the many meanings related to the “eye contact effects” by referring to these two works since they are relevant for your own study.

2) Eye-gaze stimuli are powerful stimuli that can deeply shape visual attention. Other than attentional capture and holding, they can also elicit spatial orienting, as recently summarized by a recent review paper, which could be briefly mentioned to provide a broader and more complete overview of this topic (Dalmaso, Castelli, & Galfano, 2020)

3) Lines 111-114: I wouldn’t say your task doesn’t involve conscious control, especially because you focused on fixations, which can be modulated by participants' will. Please clarify this paragraph.

4) Looking at figure 1, one face was Asian, so I’m wondering how many other non-Caucasian faces were used (assuming that your participants were all caucasian). It’s well established that race can impact face processing and face scanning. I’m thinking, for instance, about a seminal work by Blais and colleagues you might cite (Blais, Jack, Scheepers, Fiset, & Caldara, 2008). Hence, I’m wondering why you decided to include faces belonging to different ethnic groups. More details about participants’ ethnicity could be also provided (if possible).

5) I’m not an expert on eye-tracking methodology, but just wondering whether some works tested the reliability of the data you’ve collected in such a way. Referring to some methodological papers could support your approach more strongly.

6) Data analysed split into two different time windows (500 ms and 5000 ms) could sound rather post-hoc, at least to me (you only cited one of your own work, Fedrico & Bradimonte 2019, to support this approach), despite the very same results emerged in both windows. So, I guess you might be able to provide more theoretical motivations for that.

7) Did you check whether participants contracted the virus? (or even if they were positive at the time of the experiment). This should be definitely clarified.

8) Figure 3: figures showing “example” participants should be always avoided, at least to me. You should show mean data from the whole sample, otherwise please remove the pic.

References
Blais, C., Jack, R. E., Scheepers, C., Fiset, D., & Caldara, R. (2008). Culture shapes how we look at faces. PLoS ONE, 3(8). https://doi.org/10.1371/journal.pone.0003022

Conty, L., George, N., & Hietanen, J. K. (2016). Watching Eyes effects: When others meet the self. Consciousness and Cognition, 45, 184–197. https://doi.org/10.1016/j.concog.2016.08.016

Dalmaso, M., Castelli, L., & Galfano, G. (2020). Social modulators of gaze-mediated orienting of attention: A review. Psychonomic Bulletin & Review, 27(5), 833–855. https://doi.org/10.3758/s13423-020-01730-x

Senju, A., & Johnson, M. H. (2009). The eye contact effect: mechanisms and development. Trends in Cognitive Sciences, 13(3), 127–134. https://doi.org/10.1016/j.tics.2008.11.009

Reviewer 2 ·

Basic reporting

In the submitted manuscript, the authors compared gaze exploration behavior of faces that were previously labelled as: COVID-19 Free, Sick with COVID-19 and Recovered from COVID-19. The authors analyzed the eye movement data in two regions of interest, eyes and mouth, and reported that participants fixations were longer when looking at COVID-19 Free faces compared to the other two categories. This was true both for the first 500 ms of visual exploration as well as for the full exploration time of 5000 ms. Moreover, in the full exploration time window, participants fixated more the mouth region of Sick with COVID-19 and Recovered from COVID-19 faces, compared to COVID-19 Free faces. The authors suggest that the exploration behavior is indicative of a threatening perception of the COVID-19 faces (i.e. less time spent looking at the eyes) with more attention deployed on the possible vehicle of transmission (i.e. more time spent looking at the mouth in COVID-19 and Recovered from COVID-19 faces).

Overall, I think that the paper is very well written. The introduction and the discussion are exhaustive and well referenced, the method and result sections are very clear and easy to follow. The authors also provide all the raw data collected for the study.

Experimental design

The research question and the hypotheses are clearly stated at the end of the introduction section. I find the methods used in the present paper quite innovative. First the use of machine learning algorithms to create the stimuli and avoiding possible confounds related to facial features. Then, the use of an online platform allowing to reliably record eye movement patterns that, despite the spatio-temporal resolution constrains of the webcam, represent a very informative marker of people behavior. In general the method section is complete and properly described.

2.1 I have one question that I would like the authors to comment. The authors conclude that the avoidance behavior expressed with shorter fixation time on the Sick with COVID-19 faces reveal avoidance for a threatening stimulus. Do the authors expect to see a similar pattern of results if also another disease was used as immunological status? I think this would have been a good control to understand the impact of being in the middle of a pandemic on the perception of different diseases.

2.2. In the example of the stimuli used, the faces represent relatively young individuals. We know that COVID -19 affects especially the older portion of the population. From which age range were the faces generated for the study? Do the authors expect to see an effect of “age”, considering that the severity of the disease is directly linked with this factor?

Validity of the findings

3.1 The difference in mean fixation between the different conditions is very small, about 25 ms in the first 500 ms interval and about 200 ms in the 5000 ms interval. From an eye movement perspective, 25 ms is about the time of saccade and I am a bit skeptical that it is enough to gather relevant information about the stimulus to support a difference between the two conditions. The 200 ms difference is a more reasonable duration, basically the mean duration of a fixation when looking at a stimulus. How do the authors interpret the magnitude of the effect they report? Are 25 ms really enough to acquire information and to lead to the avoidance of the stimulus? Moreover, the temporal resolution of the webcams used for the experiment is of 20Hz, that is 50 ms. This gives only 10 samples in the 500 ms time interval. This methodological aspects reduces my confidence in the 25 ms difference reported by the authors.

Additional comments

Minor comments.

In general, I would suggest to change the measure to "dwell time" rather than fixation time. Fixation duration usually refers to the duration of the single fixation, that I think you can not derive from your recording system.

For the plot, I would suggest to maybe use bars rather than lines, since “eyes” and “mouth” are two categorical factors and there’s no need to link them with a line. Also, it is not reported what the error bars represent in the plot.

Line 78 and elsewhere: The “JOHNSON” reference is formatted in capital.
Line 179: This reference seems to have the wrong formatting: “P Slovic, 198”
The numbering of the figures is weird. Figure 5 seems to be presented before Figure 2 and 3.

Reviewer 3 ·

Basic reporting

Line 48-50 it is not the depigmentation to be crucial but the relative contrast between sclera and iris (see Perea-Garcia et al, 2019: https://www.pnas.org/content/116/39/19248). Hence other primates are able to follow the gaze of a cospecific.

Experimental design

Line 304: The scree test could be unrelieble. Please use Horn’s parallel analysis, "the method of consensus in the literature on empirical methods for deciding how many components/factors to retain" (Dinno, 2009).

Validity of the findings

From the results it is not clear the role of the mouth as a possible threat. This is demonstrated by the fact that there is a significant correlation between severity and fixations of the mouth in the recovered-from-COVID-19 face condition but not in covid-19 face condition, it is puzzling this absence of a significant correlation.
A possible solution, in my opinion, is to add open mouth faces in the experimental design. Following authors' hypothesis an open mouth, representing a more dangerous threat, should collect more fixations then a closed mouth. This latter condition could solve what I think is the major weak point of this work.

Additional comments

I found the paper very interesting but affected by a major weakness. In order to make the conclusion more sound I think authors must improve their experimental design as suggested in section 3.

---

## Round 0.2 · Minor Revisions

Dear Authors, Please tend to the minor corrections suggested by one peer reviewer.Thanks

Reviewer 1 ·

Basic reporting

I'm happy with the revised version of the manuscript.

Experimental design

I'm happy with the revised version of the manuscript.

Validity of the findings

I'm happy with the revised version of the manuscript.

Additional comments

I'm happy with the revised version of the manuscript.

Reviewer 2 ·

Basic reporting

no comment

Experimental design

no comment

Validity of the findings

no comment

Additional comments

The authors responded to all my questions. The corrections provided by the authors improved the readability and the clarity of the manuscript.

Reviewer 3 ·

Basic reporting

The paper has been improved as suggested

Experimental design

Is acceptable given the the measure put in place to prevent the spread of the virus.

Validity of the findings

The authors improved their conclusions

Additional comments

I appreciated the work done by the authors to improve the paper. I understand their problems in managing a further control condition (I.e an open mouth or a much more simpler case consisting of a mouth covered by a protective mask). Given the resolution of the webcam the authors used (20hz), a problem also raised by another reviewer, I suggest the authors to stress this limitation in all the captions in which ms are reported, in fact the resolution is not 1ms but rather 50ms! Furthermore, the second part of the title of the paper can be misleading, clearly it is not covid19 per se that changes the way we look but instead the risk to acquire the infection.

---

## Round 0.3 · accepted · Accept

Congratulations your publication is accepted!

Reviewer 3 ·

Basic reporting

The paper has been improved as suggested.

Experimental design

The paper has been improved as suggested.

Validity of the findings

The paper has been improved as suggested.

Additional comments

The paper has been improved as suggested.